# Genetic reprogramming of human amniotic cells with episomal vectors: neural rosettes as sentinels in candidate selection for validation assays

Patricia G. Wilson[1] and Tiffany Payne[2]

[1] Institute for Regenerative Medicine, Wake Forest School of Medicine, Winston Salem, NC, USA
[2] Wake Forest University, Winston Salem, NC, USA

## ABSTRACT

The promise of genetic reprogramming has prompted initiatives to develop banks of induced pluripotent stem cells (iPSCs) from diverse sources. Sentinel assays for pluripotency could maximize available resources for generating iPSCs. Neural rosettes represent a primitive neural tissue that is unique to differentiating PSCs and commonly used to identify derivative neural/stem progenitors. Here, neural rosettes were used as a sentinel assay for pluripotency in selection of candidates to advance to validation assays. Candidate iPSCs were generated from independent populations of amniotic cells with episomal vectors. Phase imaging of living back up cultures showed neural rosettes in 2 of the 5 candidate populations. Rosettes were immunopositive for the Sox1, Sox2, Pax6 and Pax7 transcription factors that govern neural development in the earliest stage of development and for the Isl1/2 and Otx2 transcription factors that are expressed in the dorsal and ventral domains, respectively, of the neural tube *in vivo*. Dissociation of rosettes produced cultures of differentiation competent neural/stem progenitors that generated immature neurons that were immunopositive for $\beta$III-tubulin and glia that were immunopositive for GFAP. Subsequent validation assays of selected candidates showed induced expression of endogenous pluripotency genes, epigenetic modification of chromatin and formation of teratomas in immunodeficient mice that contained derivatives of the 3 embryonic germ layers. Validated lines were vector-free and maintained a normal karyotype for more than 60 passages. The credibility of rosette assembly as a sentinel assay for PSCs is supported by coordinate loss of nuclear-localized pluripotency factors Oct4 and Nanog in neural rosettes that emerge spontaneously in cultures of self-renewing validated lines. Taken together, these findings demonstrate value in neural rosettes as sentinels for pluripotency and selection of promising candidates for advance to validation assays.

Corresponding author
Patricia G. Wilson,
pgwilson@wakehealth.edu

## INTRODUCTION

Genetic reprogramming offers unprecedented opportunities for regenerative medicine (*Robinton & Daley, 2012*; *Trounson, Shepard & DeWitt, 2012*; *Yamanaka, 2012*). Genetic

reprogramming of fetal cells in amniocentesis samples is a feasible path to fetus-specific iPSCs for testing the efficacy of pharmaceuticals and for postnatal therapies. From a practical viewpoint, reprogramming of autologous fetal cells for translational use is less likely in the foreseeable future than use of immunologically compatible iPSCs from allogenic sources that have been reprogrammed and manufactured with GMP compliant standards (*Turner et al., 2013*). From this standpoint, fetal cells in amniotic fluid are attractive because they are among the youngest cells available with minimally invasive procedures.

Amniotic cells are unique among targets for genetic reprogramming in that they are drawn from a fluid-filled reservoir rather than a vascularized tissue. Amniocentesis samples contain a mixture of cells that are sloughed from exposed fetal and placental surfaces into amniotic fluid (*Maguire et al., 2013*; *Wilson et al., 2012*). Although amniotic cells are most widely known as stromal cells (*Murphy & Atala, 2013*), fetal skin and placental membranes expose the largest surface area to amniotic fluid (*Dobreva et al., 2010*) and these epithelia are likely significant contributors of cells to amniocentesis samples (*Jezierski et al., 2010*). Amniotic fluid is primarily derived by flow from the placenta and fetal lungs into the amniotic sac (*Brace, 1997*) and it is composed mainly of water with some electrolytes and urea from fetal urine (*Underwood, Gilbert & Sherman, 2005*). A small subset of cells in amniocentesis samples can proliferate in serum-containing media *ex vivo*; clonal analysis of independent amniocentesis samples indicate that the vast majority of cells do not proliferate (*Wilson et al., 2012*). Amniotic cell cultures show diversity within and among cell populations (*Wilson et al., 2012*) that may reflect genetic differences and sampling as well as congenital influences such as placental function, environmental toxins, maternal hormones or simply the length of time that founder cells remained in amniotic fluid before *ex vivo* culture. The impact of the gestational environment on amniotic cells is not well established and likely to vary among cells, but it is clear that these cells have a finite lifespan in culture and eventually undergo senescence (*Wolfrum et al., 2010*).

Genetic reprogramming can be incomplete and costly in time and resources as a result. Methods to quickly identify promising candidates can reduce this investment and differentiation potential is a logical metric. Neural differentiation of PSCs has been well characterized and is manifested in living cultures by assembly of neural rosettes (*Elkabetz et al., 2008*; *Liu & Zhang, 2011*; *Wilson & Stice, 2006*; *Zhang, 2006*), radial arrangements of polarized neuroepithelial stem cells, designated here as neural stem/progenitors (NSPs). Rosette assembly and differentiation recapitulates well characterized pathways of neurodevelopment *in vivo* (*Cohen, Briscoe & Blassberg, 2013*). The transition of PSCs through specification of neuroepithelial stem cells and restriction of cell fate to region-specific subtypes can be traced by spatial and temporal expression of transcription factors that govern neural development *in vivo* (*Elkabetz & Studer, 2008*; *Wilson & Stice, 2006*). Rosette assembly has primarily been used to characterize neural differentiation in established PSC lines (*Elkabetz & Studer, 2008*; *Shin et al., 2006*), but it is widely recognized and recently documented that neural rosettes emerge spontaneously in cultures of self-renewing PSCs as (*Malchenko et al., 2014*).

Amniotic cells have been reprogrammed with viral vectors, including both integrating (*Anchan et al., 2011*; *Fan et al., 2012*; *Galende et al., 2010*; *Ge et al., 2012*; *Li et al., 2009*; *Li et al., 2013*; *Liu et al., 2012*; *Lu et al., 2011*; *Wolfrum et al., 2010*; *Ye et al., 2010*) and nonintegrating viral systems (*Jiang et al., 2014*), that efficiently deliver reprogramming transgenes. Leaky or reactivated expression of integrated vector transgenes can hinder differentiation and induce tumors *in vivo* (*Malik & Rao, 2013*; *Mostoslavsky, 2012*; *Rao & Malik, 2012*), blocking clinical translation as a result. Nonintegrating vectors circumvent this barrier (*Mostoslavsky, 2012*) and transgene-free iPSCs have been derived from stromal cells in amniotic fluid using a commercial source of nonintegrating Sendai viral vectors (*Jiang et al., 2014*). Nonintegrating episomal vectors for reprogramming are attractive because they are easily accessible and cheaply amplified with well-established methods that are used in most research labs (*Mostoslavsky, 2012*). Vectors have improved since their introduction, but reprogramming efficiency of episomal systems remains lower than that of viral systems.

Our previous work isolated a collection of independent amniotic cell cultures in an effort to define the diversity in amniotic cell populations (*Wilson et al., 2012*). Donated samples were diluted with serum containing media and directly plated in culture wares without prior centrifugation or refrigeration to minimize loss due to sample manipulation. Some samples were minimally diluted and contained a mixture of stromal and epithelial cell types on the basis of cell morphology. Other samples were similarly isolated except they were highly diluted and plated in multiwell culture wares to generate clonal populations that expanded without paracrine signaling exists in mixed cell populations. One inference of this work (*Wilson et al., 2012*) and additional unpublished results is that fewer than 15 founder cells initiate mixed cell populations. Molecular and cytological analysis of mixed cell and clonal populations showed diversity within and among populations, but stromal and epithelial cells alike shared characteristics of stromal cells, as if epithelial cells exposed to amniotic fluid, before or after entering amniotic fluid, had initiated epithelial to mesenchymal transition (EMT), a process in which epithelial cells acquire stromal cell traits (*Nieto, 2011*). A second inference is that, given that amniotic fluid is exposed principally to epithelial surfaces, is that the bulk of cells in amniotic fluid may be epithelial cells at various stages of EMT.

Here we report use of first-generation episomal vectors (*Yu et al., 2009*) to genetically reprogram independent amniotic cell populations. Given the cost in time and resources required for genetic reprogramming and the uncertainty of reprogramming stromal-like epithelial cells, our strategy was to use assembly of neural rosettes as a sentinel assay to screen and select candidates to advance for validation assays.

## MATERIALS AND METHODS

### Amniotic cell sources and nomenclature

Amniotic cell populations described herein were derived from amniocentesis samples (*Wilson et al., 2012*) that were donated with informed consent and a protocol approved by the Institutional Review Board of Wake Forest University Health Sciences (IRB#00007486).

We were blinded to age of the mother, period of gestation or the results of diagnostic tests. Following transfection of target cells and colony isolation, derivative lines were indicated as iChM5 or iChMRC.B1 candidates and designated as iPSCs only following successful validation assays. By convention the passages (p) number is indicated as an extension of the population name where relevant. iChM5A and iChM5B populations are referred to collective as iChM5 derivatives for simplicity and likewise, independent candidate lines that were derived from ChMRC.B1 cells are referred to as independent iChMRC.B1 derivatives.

## Somatic cell culture

Amniotic cells and HEK293 cells were maintained in DMEM15% (DMEM supplemented to 15% fetal bovine serum (FBS), 1% L-glutamine and 1% penicillin/streptomycin solution). Cells were routinely maintained on culture wares pretreated with 1:100 dilution of growth factor reduced matrigel (BD Biosciences). All media components in this work were obtained from Life Technologies unless stated otherwise.

## PSC cell culture

The H9 (WA09) line of human embryonic stem cells (hESCs) (*Thomson et al., 1998*) and iPSC lines were maintained and/or established with a feeder-dependent culture system and standard hESC media supplemented with 1% penicillin/streptomycin solution on mitomycin-C inactivated mouse embryonic fibroblasts (MEFs) as recommended by the National Stem Cell Bank (NSCB, Madison WI). MEFs were generated from 13-day old CF-1 embryos (Charles River, Inc) and following expansion and mitomycin-C treatment, MEFs were washed extensively with Dulbecco's phosphate buffered saline (DPBS; Life Technologies), harvested with Accutase (Life Technologies) and replated in MEF media on culture wares near $2 \times 10^5$ cells/cm$^2$ for immediate use or cryopreserved with standard methods after 24 h recovery. Conditioned hESC media was prepared by culture of inactivated MEFs in hESC media without bFGF for 24 h, supplemented with 4 ng/ml bFGF and filtered sterilized before use. Feeder-free cultures were maintained in MEF-conditioned hESC media, mTeSR-1 (StemCell Technologies) or Essential 8 (Life Technologies) media. Passaging of PSCs cultured on MEF feeders or in MEF-conditioned media was done by manual microdissection of optimal undifferentiated colonies with a fire-polished glass pipette using a dissecting microscope. Feeder-free cultures were passaged with EDTA as described (*Beers et al., 2012*). The ROCK inhibitor Y27632 (Tocris) was routinely added at 5 µM/ml media for 24 h post-passage.

## Genetic reprogramming

The episomal vectors (Addgene, Inc.) that were used in this work are described in Table 1. Episomal vectors were amplified in Top10 bacteria with antibiotic selection in standard Luria Broth and extracted with DNAeasy Kits (Qiagen, Inc) with good recovery of DNA. In each experiment $\sim 8 \times 10^5$ target cells were seeded at subconfluent densities $\sim 1.4 \times 10^3$ cells/cm$^2$ and transfected the following day with pooled plasmid combinations in equimolar ratios ($\sim 0.2$ µg DNA/cm$^2$) with Fugene HD (Promega, Corp.) 0.15 µl/µg DNA at 8 to 12 h intervals for a total of 3 transfections. Transfected

**Table 1 Episomal vectors[a] and vector combinations.** This table described the vectors used in this study, including size of each vector, combinations that were used and structure of the transgene cassettes.

| Episomal vector[b] | Kb[c] | Combinations[d] | Cassette 1[e] | Cassette 2 | Casette 3 |
|---|---|---|---|---|---|
| pEP4 E02S CK2M EN2L | 20 | 2 | EF1α-O-2A-S | CMV-K-2A-Mf | EF1α-N-2A-L |
| pEP4 E02S ET2K | 18 | 2, 3 | EF1α-O-2A-S | EF1α-T-2A-K | |
| pEP4 E02S EN2K | 16 | 3 | EF1α-O-2A-S | EF1α-N-2A-K | |
| pCEP4 M2L | 13 | 3 | CMV-M-2A-L | | |

**Notes.**

[a] Plasmids contain 1 to 3 bicistronic cassettes in pIRES vectors with self-cleaving 2A peptides located between transgenes. Transgene cassettes are expressed under control of the EF1α or CMV promoters as indicated. *Yu et al. (2009)* successfully reprogrammed fetal dermal fibroblasts with 3 combinations (#) of plasmids: #4:1,2; #19:2,5,6; and #6:2,4,5 as designated.

[b] Reprogramming transgenes include Oct4 (O), Sox2 (S), Klf4 (K), cMyc (M), Lin28 (L), and Nanog (N), SV40 Large T antigen (T).

[c] Vector size.

[d] The 2-vector (2) and 3-vector (3) combinations correspond to combinations #4 and #6 (*Yu et al., 2009*).

[e] Bicisitronic cassettes listed in the order in plasmid.

cells were maintained in DMEM 15% for ∼4 days and then switched to MEF conditioned hESC media supplemented with 2.5 mM valproic acid (Sigma-Aldrich) for ∼2 weeks after colonies appeared. Independent populations of ChMRC.B1p3 cells were transfected with the 3-vector combination and 7 to 9 colonies were recovered from each population. A single representative colony was selected from each and maintained separately as iChMRC.B1A, iChMRC.B1C, and iChMRC.B1E candidate populations. A population of ChM5p10 cells was transfected with the 2-vector combination, but the population became highly confluent in hESC media within 2 weeks and potential colonies were difficult to identify. The transfected population was passaged with Accutase and replated on MEF feeders. hESC-like colonies emerged within 2 weeks, optimal colonies were pooled and maintained as the iChM5A candidate population. Transfected ChM5p12 cells were maintained for 4 days in growth media, treated with Accutase and passaged to MEF feeders as separate populations; a single hESC-like colony was recovered from one population of transfected cells and maintained as the iChM5B candidate population. Optimal hESC-like candidate colonies and control H9 hESC colonies were passaged as needed to maintain healthy cultures.

## Neural differentiation

Following the first manual passage of candidate colonies from MEF feeders, residual colony fragments in the primary culture plate were maintained in conditioned hESC media without bFGF for 3 to 5 days to allow colony expansion and then switched to regular hESC media to encourage spontaneous differentiation as the MEF feeders age and pluripotency of the expanding population by the absence of bFGF in hESC media. Rosettes were manually isolated as they emerged and passaged in hESC media to matrigel-treated cover slips for immunostaining. Long term cultures of neural progenitors/stem cells (NSPs) were established as described (*Shin et al., 2006*); neural rosettes were serially passaged for 2 or 3 times to enrich for rosettes before dissociation with Accutase and

population expansion. Rosette-derived NSP cultures and a commercial source (Millipore) immortalized human midbrain NSPs (hVMNSPs) were maintained in ReNcell NSC Maintenance Media (Millipore) supplemented with 20 ng/ml bFGF and 20 ng/ml EGF or a proliferation media (1:1 mix of DMEM/F-12 and Neurobasal media, 1% L-glutamine and 1% penicillin/streptomycin solution, 0.5 X B27, 0.5X N2, 20 ng/ml bFGF and 20 ng/ml EGF) as described (*Brace, 1997*). Differentiation of NSPs was induced by withdrawal of bFGF and EFG from proliferation media. Rosette collections and NSPs were cryropreserved in proliferation media supplemented with 10% DMSO with standard methods. Addition of ROCK inhibitor greatly improved survival at thaw.

## PCR detection of transgene and vector sequences

Total cellular DNA was isolated with GenePure (Qiagen) or QiaAmp DNA Mini (Qiagen) kits and treated with RNAse to remove RNA. Transgenes or endogenous genes were amplified in reactions containing 100 ng genomic DNA or <1 ng plasmid DNA with GC-rich polymerase (Life Technologies) in 1X Buffer A, 3 μl of Enhancer and 250 nM of oligonucleotide primers (Table 2) with touchdown cycling conditions: 1 cycle [95 °C for 10 min], 2 cycles [95 °C for 1 min, 64 °C for 1 min, 72 °C for 1 min], 2 cycles [95 °C for 1 min, 62 °C for 1 min, 72 °C for 1 min], 2 cycles [95 °C for 1 min, 60 °C for 1 min, 72 °C for 1 min], 35 cycles [95 °C for 1 min, 58 °C for 1 min, 72 °C for 1 min] and 1 cycle [72 °C 10 min].

## Transcript analysis

Total cellular RNA was isolated with RNAeasy kits (Qiagen) and contaminating DNA was removed by DNAse treatment. RNA was converted to cDNA using SuperScript First-Strand Synthesis System (Life Technologies) and 1 μl of 1:4 dilution of cDNA in water was amplified in each reaction. Transcript levels in Fig. 8A were assayed with QuantiTect Syber Green primer assays (Qiagen) with the exception of cMyc primers (Table 2) with FastStart Universal SYBR Green Master Mix (Roche/Life Technologies). Transcript levels in Fig. 8B were established with TaqMan assays using TaqMan® Gene Expression Master Mix (Life Technologies). Amplification rates (Ct values) of cDNA were assayed in more 2 replicates for each gene. The mean (AVG) and standard error (SE) was calculated with the Descriptive Statistics tool in Excel and normalized to $\beta$-glucuronidase (GUSB) with the $\Delta\Delta$Ct method described in Applied Biosystems/Life Technologies technical resources.

## Bisulfite sequencing

Genomic DNA was processed with an Epitect kit (Qiagen) as directed by the vendor. Amplification products were generated with primers that were specific to converted DNA (Table 2), purified with a Qiaquick PCR purification kit and cloned with a TOPO-TA PCR4 cloning kit (Life Technologies). Plasmid DNA was purified with QIAprep Spin Miniprep kits (Qiagen) or EconoSpin columns (Epoch) and sequenced directly or the vector inserts were first amplified with M13 primers using High Fidelity EcoDry PCR mix (Promega, Corp.) as follows: 95 °C for 10 min, 40 cycles (95 °C for 15 s, 54 °C for 30 s and 68 °C for 30 s), 68 °C for 10 min. Amplification products were column-purified and sequenced directly (Operon or Genewiz). Data was imported into the SeaView graphical software

**Table 2 Primers.**[a] This tables describes all of the primers, other than Qiagen kit contents, that were used in our study.

| PCR/RTPCR[b] | Size bp | Primer | 5′ → 3′ |
|---|---|---|---|
| TgOct4 | 657 | Oct4SF1 | AGTGAGAGGCAACCTGGAGA |
| | | IRES2SR | AGGAACTGCTTCCTTCACGA |
| TgNANOG | 732 | NanogSF2 | CAGAAGGCCTCAGCACCTAC |
| | | IRES2SR | AGGAACTGCTTCCTTCACGA |
| TgSV40LT | 491 | SV40TSF1 | TGGGGAGAAGAACATGGAAG |
| | | RES2SR | AGGAACTGCTTCCTTCACGA |
| TgSox2 | 534 | IRES2SF1 | ACCAGCTCGCAGACCTACAT |
| | | SV40pAR | CCCCCTGAACCTGAAACATA |
| TgLIN28 | 447 | Lin28-SF1 | AAGCGCAGATCAAAAGGAGA |
| | | SV40pAR | CCCCCTGAACCTGAAACATA |
| TgKLF4 | 401 | KLF4SF1 | CCCACACAGGTGAGAAACCT |
| | | SV40pAR | CCCCCTGAACCTGAAACATA |
| TgEBNA1 | 666 | pEP4SF2 | ATCGTCAAAGCTGCACACAG |
| | | pEP4SR2 | CCCAGGAGTCCCAGTAGTCA |
| TgOriP | 544 | pEP4-SF1 | TTCCACGAGGGTAGTGAACC |
| | | pEP4-SR1 | TCGGGGGTGTTAGAGACAAC |
| eOct4 | 113 | Oct4-F2 | AGTTTGTGCCAGGGTTTTTG |
| | | Oct4R2 | ACTTCACCTTCCCTCCAACC |
| eGAPDH | 152 | GAPDHF | GTGGACCTGACCTGCCGTCT |
| | | GAPDHR | GGAGGAGTGGGTGTCGCTGT |
| ecMyc[c] | 284 | cMycF | GCCACAGCATACATCCTGTCCGTCCAAGC |
| | | cMycR | CCAAAGTCCAATTTGAGGCAGTTTAC |
| Bisulfite sequencing | | | |
| OCT4_2 | −2609 to −2417 | OCT4_2F | ATTTGTTTTTTGGGTAGTTAAAGGT |
| | | OCT4_2R | CCAACTATCTTCATCTTAATAACATC |
| OCT4_4 | −2136 to −1721 | OCT4_4F | GGATGTTATTAAGATGAAGATAGTTGG |
| | | OCT4_4R | CCTAAACTCCCCTTCAAAATCTATT |
| OCT4_6 | −567 to −309 | OCT4_6F | TAGTTGGGATGTGTAGAGTTTGAGA |
| | | OCT6_2R | TAAACCAAAACAATCCTTCTACTCC |

**Notes.**

[a] Primer sequences for PCR (*Freberg et al., 2007*) and bisulfite sequencing (*Yu et al., 2009*).

[b] Transgene (Tg) and endogenous (e) genes on vectors and chromosomes, respectively.

[c] Primers for RTPCR endogenous cMyc (*Kim et al., 2009*). Other transcript profiles used EpiTect Assays (Qiagen).

program for alignment and analysis. The full set of DNA sequencing files are available at FigShare: http://dx.doi.org/10.6084/m9.figshare.1153969.

## Imaging and immunocytochemistry

Cells were cultured in multiwell tissue culture plates on cover glass or in multiwall chamber slides that were pretreated with 1:100 dilution of growth factor reduced matrigel. Samples were fixed and immunostained as described (*Wilson et al., 2012*) with antibodies tabulated in Table 3. Wide-field images were captured with ImagePro software using a QImaging CCD camera mounted on a Leica upright microscope. Immunostaining

**Table 3  Antibodies.** Here we compile a list of the antibodies used in our study, including antigem, host species, monoclonal vs polyclonal, dilution, vendor catalogue number where available.

| Antigen | Host | Dilution[a] | Cat. Number | Vendor |
|---|---|---|---|---|
| Oct3/4 | mouse | 1:300 | sc-5279 | Santa cruz |
| Sox2 | mouse | 1:200 | MAB4343 | Chemicon/Millipore |
| Nanog | rabbit | 1:500 | 4903S | Cell signaling |
| Sox1 | goat | 1:500 | sc-17317 | Santa cruz |
| Pax6 | mouse | 1:15 | PAX6 | DHSB[b] |
| Pax7 | mouse | 1:15 | PAX7 | DHSB |
| Otx2 | goat | 1:500 | AF1979 | R&D systems |
| Islet1/2 | mouse | 1:50 | 39.405F7 | DHSB |
| Eg5 | rabbit | 1:500 | NB500-181 | Novus biologicals |
| nestin | rabbit | 1:500 | ab5968 | abCAM |
| vimentin | goat | 1:200 | sc-7557 | Santa cruz |
| GFAP | rabbit | 1:500 | ab7779 | abCAM |
| $\beta$III-Tubulin | mouse | 1:1000 | MAB1195 | R&D systems |
| SSEA5 | mouse | 1:500 | ab3355 | abCAM |
| Tra-1-81 | mouse | 1:500 | 560793 | BD biosciences |

**Notes.**

[a] Monoclonal antibodies were stored in small aliquots at $-80\,^{\circ}\mathrm{C}$ and working stocks were maintained undiluted at $4\,^{\circ}\mathrm{C}$; all other antibodies were maintained in 40% glycerol at $-20\,^{\circ}\mathrm{C}$ as working stocks or $-80\,^{\circ}\mathrm{C}$ for long term storage.

[b] Developmental Studies Hybridoma Bank.

was repeated in at least 2 technical replicates and in more than 3 independent trials for each marker/combination tested. The images shown throughout this manuscript are representative; our conclusions were based on at least 3 fields of view for each replicate and inspection of more than 500 cells for detection of each antigen. Virtually all experiments were done in parallel with positive and negative controls, typically H9 hESCs, parental ChM5 cells or HEK293 as appropriate for the antigen.

## RESULTS

### Target amniotic cell populations and nomenclature

Amniotic cell populations were derived from amniocentesis samples (*Wilson et al., 2012*) that were donated with informed consent and a protocol approved by the Institutional Review Board of Wake Forest University Health Sciences (IRB#00007486). We were blinded to age of the mother, period of gestation or the results of diagnostic tests. Derivative amniotic cell populations were designated ChM populations, referencing the Christopher Moseley Foundation as the funding source, and a unique identifier. Each mixed cell population was assigned a number and each clonally derived line was assigned an alphanumeric identifier to reflect the amniocentesis sample and the 12-well plate number (if multiple) and an extension (.) that corresponded to the well address of the clone (*Wilson et al., 2012*). The passage (p) number where relevant is indicated by convention as an extension of the population name (p#).

Reprogramming targets were selected to reflect the range of cell types in amniocentesis samples and proliferation characteristics that we considered to be important to the

efficiency of reprogramming. The ChM5 mixed cell population was highly enriched for fibroblast-like stromal cells and cell proliferation continues in confluent cultures, generating dense stratified cell layers (*Wilson et al., 2012*). The ChMRC.B1 clonal population of stromal-like epithelial cells, designated here as epithelial for simplicity, continues to expand in subconfluent cultures, but shows contact inhibition of proliferation in confluent cultures (*Wilson et al., 2012*), verified by the absence of mitotic figures by immunofluorescence analysis of chromosomes and spindle microtubules (data not shown).

## Episomal vector maintenance and Oct4 immunostaining controls

Reprogramming used combinations of 2 or 3 first generation episomal vectors (Table 1) that collectively encoded the four Yamanaka factors Oct4, Sox2, Klf4, cMyc (*Takahashi et al., 2007*) as well as Nanog, Lin28 and the Large T antigen of SV40 (*Yu et al., 2009*). Preliminary experiments showed efficient transfection of HEK293 cells with Fugene-HD and correlated maintenance of vector sequences with immunostaining of Oct4 (Fig. 1). Transfected HEK293 cells were serially passaged; one-third of the population was plated for continued culture, total DNA was extracted from one-third for polymerase chain reaction (PCR) with published primers (Table 2) and one-third was fixed and immunostained for Oct with a well characterized antibody (Table 3). PCR analysis did not detect the Oct4 transgene in HEK293 cells before transfection but Oct4 was detected in serial passages 1 through 4 (Fig. 1A), showing that the episomal vectors were present, but unstably maintained in cell populations that were cultured in serum containing media. Primers and antibodies used throughout this work are described in Tables 2 and 3, respectively.

Immunostaining of transfected populations showed a high frequency of Oct4 positive cells following transfection, but the loss of virtually all immunopositive cells by passage 5 (Fig. 1B). Haemacytometer-based counts of Oct4 positive cells showed a decrease from 5% to 0.5% over the same 5 sequential passages, correlating immunodetection of Oct4 (Fig. 1B) with PCR amplification of the Oct4 transgene (Fig. 1A). Immunostaining of targeted parental populations (Fig. 1C) with the same monoclonal antibody against Oct4 failed to show expression in more than 3 experiments and inspection of more than 500 cells each. We infer from these findings that the targeted parental ChM5 and ChMRC.B1 cell populations did not express the endogenous Oct4 at detectable levels.

## Recovery and preliminary screen of candidate iPSC colonies

The efficiency of chemical transfection of amniotic cell targets was low; less than 5% of cells were immunopositive for Oct4 at 48 h post-transfection. Subconfluent cultures of $\sim 8 \times 10^5$ cells were serially transfected every 8 to 12 h for 3 transfections in order to increase the number of transfected cells. ChMRC.B1p6 cells were transfected with the 3-vector combination (Table 1) in 3 separate populations and 7 to 9 candidate colonies were generated in each population. A representative colony from each population was identified by inspection and manually isolated by colony microdissection with a fire-polished pulled glass pipette and expanded independently as the iChMRC.B1A, iChMRC.B1C, and iChMRC.B1E candidate populations. ChM5p10 and ChM5p12 cells were transfected with the 2- and 3-vector combination, respectively (Table 1); multiple

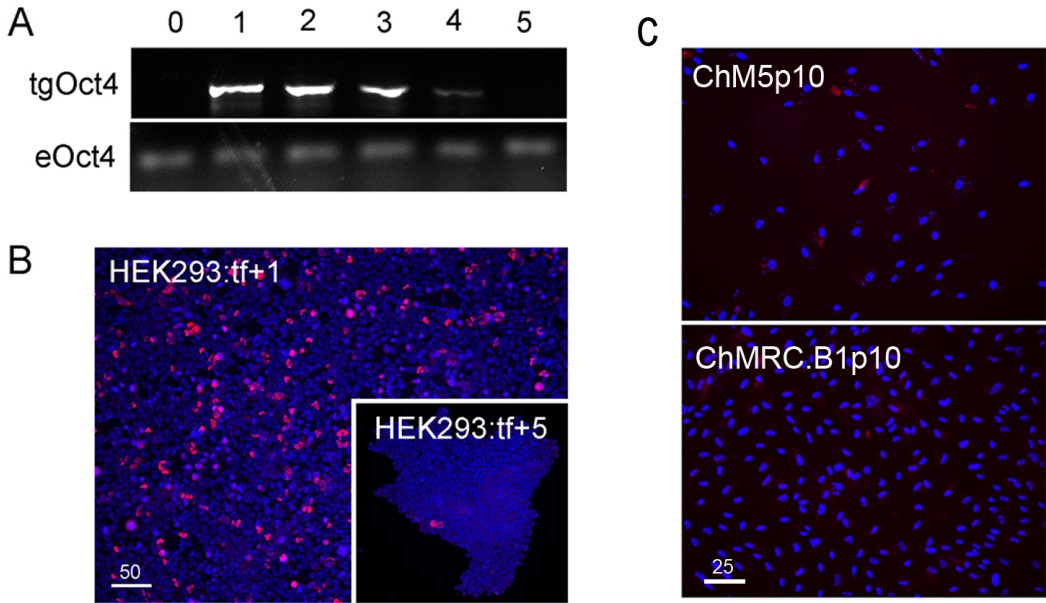

**Figure 1 Validation of vectors and immunospecificity of Oct4 in control and targeted cells.** (A) *PCR analysis.* Amplification of the vector-borne Oct4transgene (tgOct4) and endogenous chromosomal Oct4 (eOct4) in nontransfected control HEK293 cells (0) and HEK293 cells transfected with 2-vector combination of the pEP4 E02S CK2M EN2L and pEP4 E02S ET2K plasmids at passages 1 through 5 in serum containing media as indicated. Transfected populations were serially passaged, counted with a haemocytomer at each passage and a portion of each population at each passage was used for DNA isolation, immunostaining and seeding new cultures with defined cell numbers. (B) *Immunostaining of Oct4.* The first (HEK293:tf+1) and last passage (HEK293:tf+5) of transfected cells showed 5% and 0.5%, respectively, of the cells were immunopositive for Oct4. These findings suggested that episomes were not efficiently replicated and were rapidly lost during expansion of HEK293 populations in DMEM15% media. (C) Immunostaining of Oct4 in nontransfected targeted populations. Here we show representative fields ofnontransfected ChM5p10 and ChMRB.B1p10 cells that were immunostained for Oct4, using the same monoclonal antibody used in staining HEK293 cells. Repeated independent trials ($n > 3$) failed to show nuclear localized staining.

colonies were generated by transfection of ChM5p10 cells and a single colony was recovered by transfection of ChM5p12 cells. The ChM5p10 and ChM5p12 derivative candidate lines were designated as iChM5A and iChM5B, respectively. Given the low frequency of transfection and known inefficiency of these first generation episomal vectors (*Yu et al., 2009*), candidate iChM5A colonies maybe siblings that were derived from the same the founder. We manually isolated by colony microdissection and pooled candidate iChM5A colonies to conserve resources, reasoning that clonal populations could be established as needed.

Optimal colonies growing on feeders were identified by eye and manually recovered by colony microdissection to expand populations. Optimal colonies were defined as those similar to colonies of H9 hESCs (Fig. 2). Colonies of iChM5 derivatives were compact with well-defined edges; colonies of iChMRC.B1 derivatives were similar, but less compact. Cells in candidate colonies were small (~15 μm in diameter) in comparison to the size of parental amniotic cells (~50 μm to 150 μm in diameter), primarily due to apparent

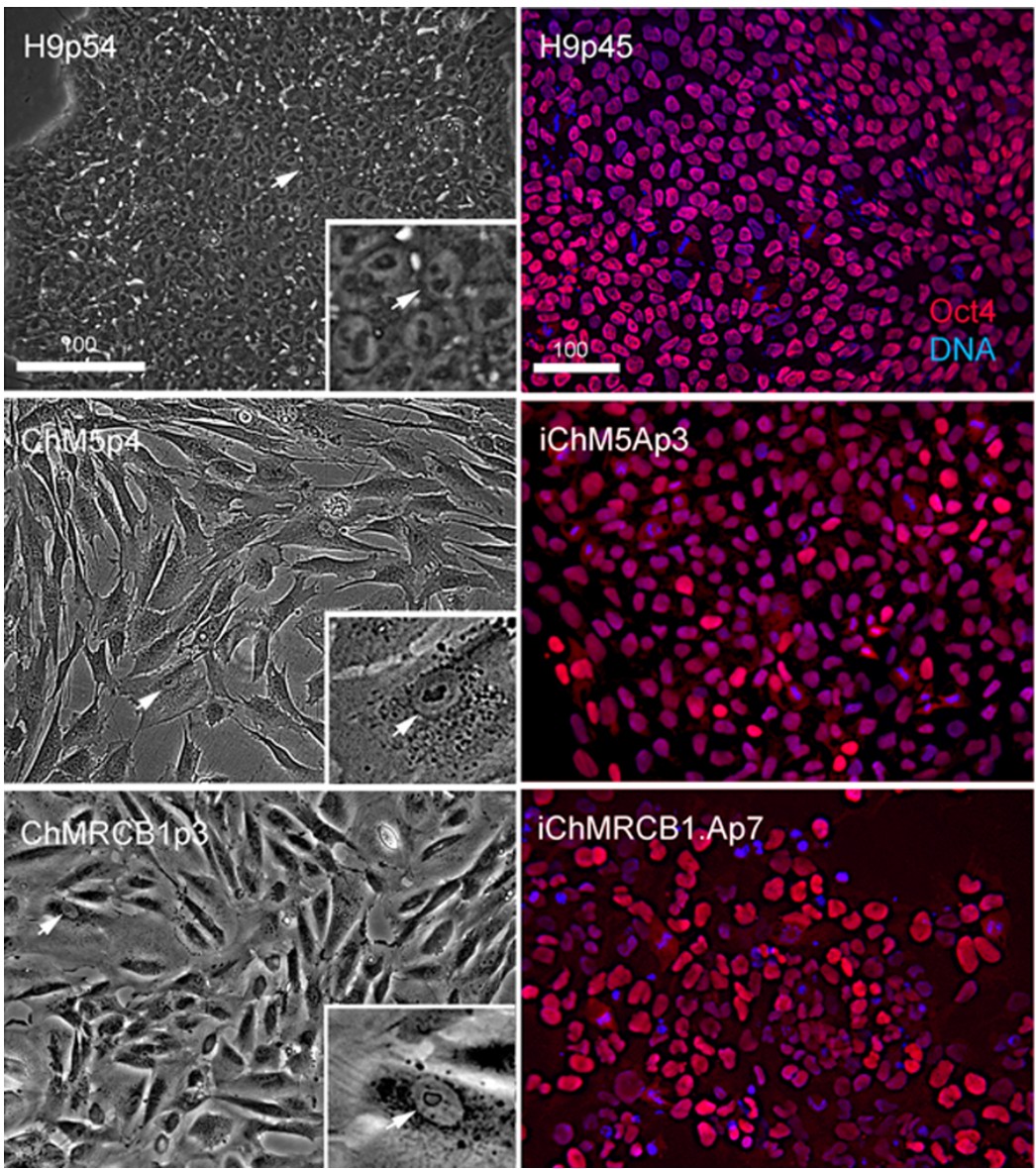

**Figure 2 Characterization of parental cells and candidate colonies.** Phase images compare the morphology of control H9p54 hESCs with parental ChM5 and ChMRCB1ChMRC.B1 cells. Inserts are magnified 3X. Note change in size due to higher area of cytoplasm in somatic cells. Magnification is identical within columns. Immunostaining of H9p45 hESCs and candidate iChM5Ap3 and iChM-RCB1ChMRC.B1Ap7 colonies for Oct4 (red) and a fluorescent DNA (blue) dye. Scale bar, 100 microns.

reduction in the amount of cytoplasm (Fig. 2). Immunostaining showed nuclear localized Oct4 expression in candidate colonies that was similar to H9 hESCs, but colonies included a subset of cells that showed obviously higher levels of Oct4 expression (Fig. 2), possibly reflecting induced expression of the endogenous Oct4 gene superimposed with Oct4 transgene expression. Taken together, these observations suggested that candidate colonies did not reflect preexisting Oct4-expressing cells. The frequencies of candidate colonies, 1 to

10 independent candidates from $\sim 8 \times 10^5$ transfected cells, was similar to previous studies using these vectors (*Yu et al., 2009*). Given the low efficiency of chemical transfection, the actual rate of colony formation may have been higher.

## Self-assembly and differentiation of neural rosettes in candidate populations

We next screened for evidence of differentiation potential to vet candidate lines for advance to more costly validations assays. Following passage of candidates to fresh feeders, sibling colony fragments were maintained in the original plate as back up cultures, initially maintained in conditioned hESC media for 3–5 days to ensure survival of the new culture and then switched to unconditioned hESC media to encourage spontaneous differentiation as feeder layers age. Rosettes did not appear in any of the backup cultures of the 3 independent lines of iChMRC.B1 candidates, despite expansion in serial passages. Neural rosettes emerged within $\sim 2$ weeks in back up cultures of iChM5A and iChM5B candidates that were indistinguishable from rosettes in control H9 hESCs (Fig. 3, Fig. S1). Rosettes were manually isolated by microdissection as they emerged in sequential backup cultures of iChM5A (p3 and p4) and iChM5B (p4 and p6) and transferred to hESC media on matrigel coated substrates for immunofluorescence analysis or to a proliferation media for cryopreservation. Immunostaining showed nuclear localization of the Sox1, Sox2, Pax6, Pax7 transcription factors (Fig. 3, Fig. S1) that regulate specification of neuroectoderm *in vivo* and the Otx2 and Isl1/2 transcription factors that determine neural subtype identity in the dorsal and ventral domains, respectively, of the neural tube (*Elkabetz & Studer, 2008*; *Hitoshi et al., 2004*; *Liu & Zhang, 2011*; *Wilson & Stice, 2006*; *Zhang, 2006*). Immunodetection of this collection of transcription factors provided strong evidence for neural identity of rosette structures. Rosettes were immunopositive for the intermediate filament proteins nestin and vimentin (Fig. S1) that are commonly used as cytoplasmic markers of neural identity, but these proteins are not exclusive to neural derivatives. All of the rosette collections that we tested showed apparent immature neurons with long axonal-like projections that were immunopositive for $\beta$III-tubulin (Fig. 3, Fig. S1). Because rosettes are unique to PSCs, we interpreted these findings as preliminary evidence for pluripotency of iChM5 candidates. Given the absence of rosettes as evidence for differentiation potential, iChMRC.B1 candidates were not pursued further here.

Neural rosettes derived from established lines of hESCs and iPSCs are a source of proliferating NSP cultures (*Elkabetz & Studer, 2008*; *Shin et al., 2006*). To test whether NSPs could be derived from iChM5 candidates, iChM5A and iChM5B candidates were differentiated toward neural lineages with an established protocol (*Shin et al., 2006*). Rosettes were manually isolated and enriched by serially passage in a proliferation media and then dissociated to generate monolayer cultures of proliferating NSPs. NSP cultures were generated from both candidates, but we focused on the NSP population that was isolated from iChM5B cultures at passage 6 (NSPB6); this population showed more than 95% of NSPs were immunopositive for Sox1 and a few $\beta$III-tubulin immunopositive immature neurons (Fig. 3). The NSPB6 population shown in Fig. 3 was maintained in

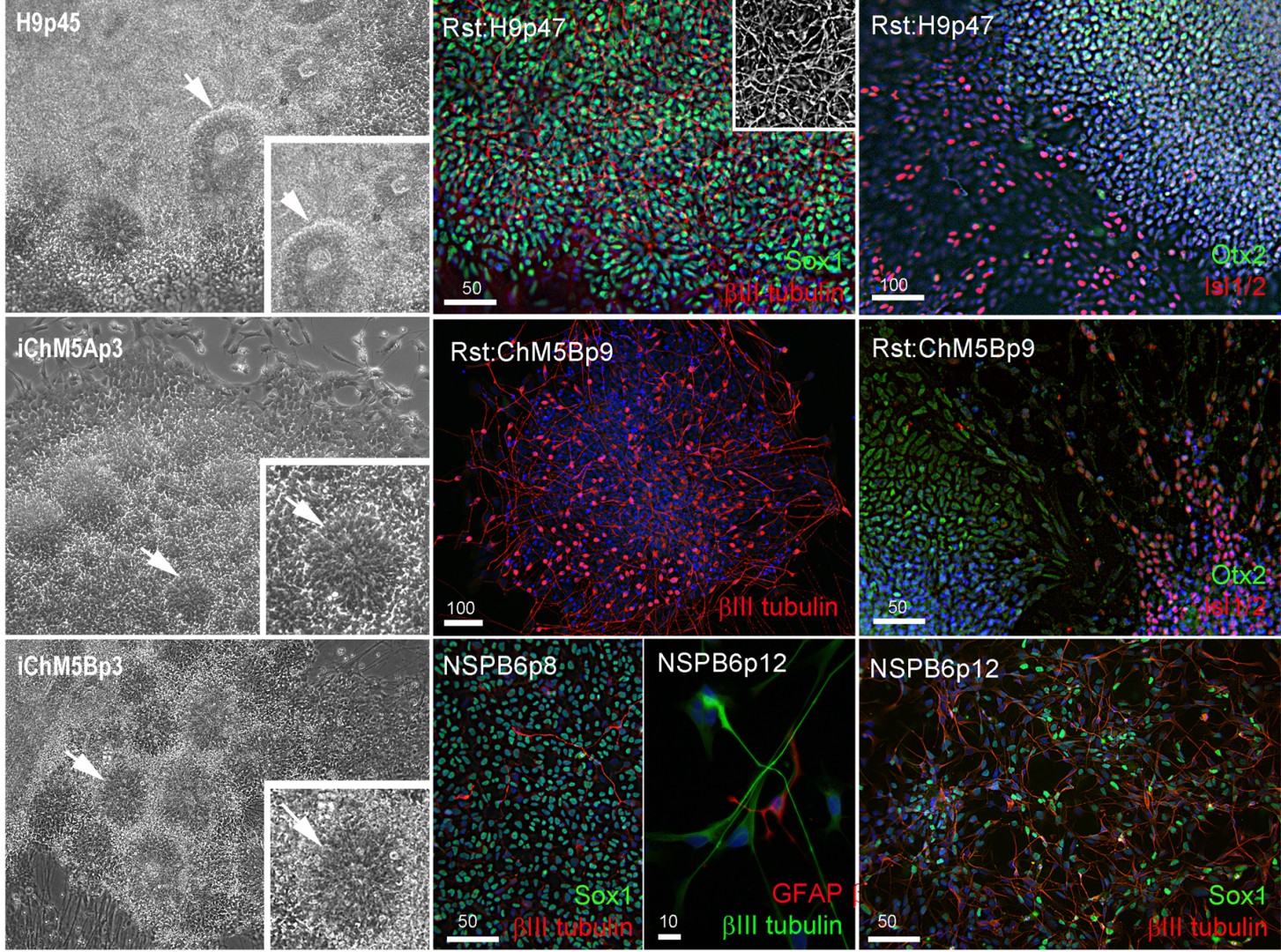

**Figure 3 Neural differentiation potential of candidate iChM5 lines.** The left column shows phase images of neural rosettes (arrows) in living cultures of pluripotent H9 hESCs and candidate iChM5 derivatives (magnification identical; insets show rosettes at 3X magnification). Color images show immunostaining of rosettes (Rst) and rosette-derived populations of neural/stem progenitors (NSPs). Shown here are NSPs derived from candidate iChM5Bp6 cultures (NSPB6) with neural markers indicated in eachpanel and chromatin stained with a fluorescent dye (blue). The grey scale inset in Rst:H9p47 cells shows immunostaining for βIII-tubulin alone to better show the density of immature neurons underlying the rosettes. These results together with immunostaining of iChM5A-derived rosettes and NSPs in Fig. S1 verify neural identity of rosette structures in backup cultures. Scale bars, in microns.

culture for more than 30 passages and produced dense mats of immature neurons that were immunopositive for βIII-tubulin (Fig. 3; Fig. S2) by withdrawal of mitogens from proliferation media to induce differentiation. Apparent glia, cells immunopositive for glia fibrillary acidic protein (GFAP), were infrequent (<1%) in all NSP populations, likely reflecting the known delay of gliogenesis relative to neurogenesis (*Wilson & Stice, 2006*). Although our analysis was not exhaustive, these findings showed derivation of differentiation-competent NSPs and provide added support for pluripotency of iChM5 derivatives and justification for advance to validation assays.

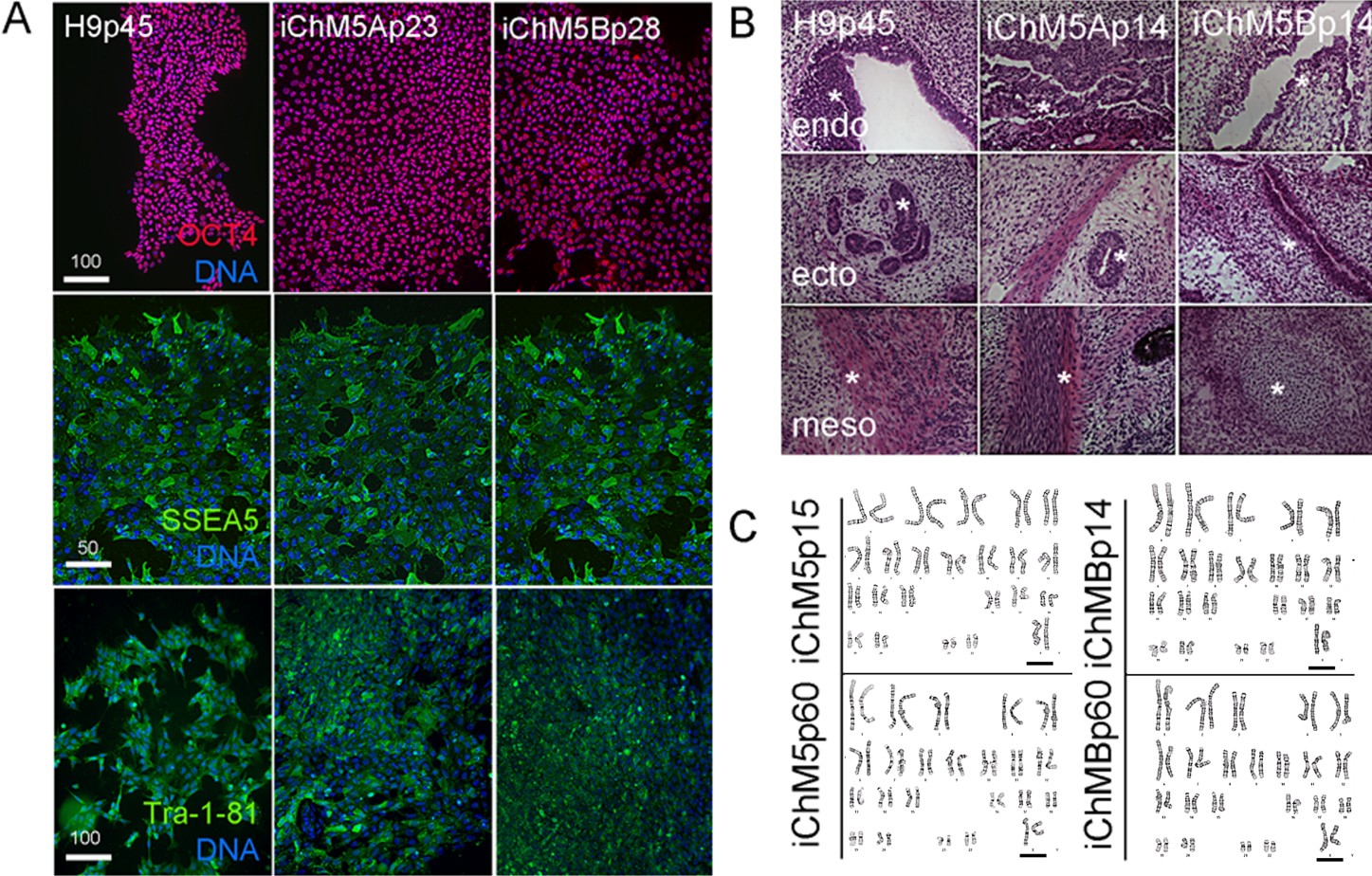

**Figure 4 Validation of pluripotency.** (A) Immunostaining of pluripotency markers. Control H9p45 hESCs, iChM5Ap23 and iChM5Bp28 cells were immunostained as indicated with antibodies against Oct4 and cell surface epitopes Tra-1-81 and SSEA5 that are associated with pluripotency. Note uniform Oct4 signal in iChM5 derivatives in comparison to early passages (Fig. 2). Scale bar, 100 microns. (B) Histochemical stains of teratomas generated with control H9p45 hESCs and iChM5Ap23 and iChM5Bp28 cells. Germ layer derivatives of endoderm (endo), ectoderm (ecto) and mesoderm (meso) in columns with examples of germ layer derivatives indicated by asterisks (*) in each teratoma. Tissue derivatives were identified with the generous help of Dr. Mark Willingham, a pathologist at Wake Forest University Health Sciences. Magnification is identical in all panels. (C) Karyotype analysis of iChM5A and iChM5B cells at early (p14) and late (p60) passages showed a normal diploid complement of chromosomes in female cells (XX, 46) without deletions greater than 5 Mb, the limits of resolution for this assay. Bar indicates XX chromsome pair. High resolution G-banded karyotype analysis was performed by the Cytogenetics Laboratory of the University of Wisconsin-Madison.

## Validation of self-renewing, karyotypically normal and pluripotent iChM5 lines

Pluripotency of iChM5 derivatives was tested with conventional validation assays. Immunostaining of iChM5Ap23 and iChM5Bp28 cultures showed expression of Oct4 (Fig. 4A), Sox2 and Nanog (see below) that was indistinguishable from expression in H9p45 hESCs. We noted that the variability in Oct4 expression that was detected in newly established populations (Fig. 1) was lost with continued culture, consistent with loss of transgene expression and/or up regulation of endogenous Oct4 expression to equivalent levels. Immunostaining showed expression of the Tra-1-81 and (*Tang et al., 2011*) SSEA-5 cell surface antigens (Fig. 4A) that are widely used as markers for pluripotency.

The gold standard for pluripotency is the capacity to generate teratomas with specialized cells and tissue derivatives of all three germ layers (*Muller et al., 2010*). Teratoma assays were used to test the developmental potential of iChM5 derivatives; injection of iChM5Ap14, iChM5Bp14 and control H9p66 hESCs in immunocompromised mice generated teratomas within 9 weeks. Histochemical stains of cryosections showed tissue derivatives of ectoderm, mesoderm and endoderm in tumors derived from iChM5 derivatives and H9 hESCs (Fig. 4B). In consultation with Dr. Mark Willingham, a trained pathologist in the Wake Forest Health Sciences, we identified ectoderm-derived pigmented skin and neural rosettes, secretory tissue typical of endoderm-derived gut, characteristic spheres of cartilage, fat cells, collagen expressing mesoderm derivatives and smooth muscle as well as rare example of striated muscle. Taken together, these findings indicated that both iChM5A and iChM5B derivatives have pluripotent developmental potential to generate tissues and specialized cells from all three germ layers.

PSCs can acquire chromosomal abnormalities during culture (*Mayshar et al., 2010*), blocking differentiation in vitro (*Wilson et al., 2007*; *Ben-David et al., 2014*) as well as use of PSCs in clinical applications. The genome integrity of iChM5 derivatives was probed with high resolution G-banded karyotype analysis by a commercial service (WiCell Cytogenetics Lab, Madison WI). This assay involves chromosomal incorporation of the intercalating dye ethidiun bromide and induced mitotic arrest with a microtubule poison that disassembles spindles, blocking cell cycle progression (www.wicell.org). Cells are fixed and prepared for digital analysis of chromosomes in mitotic figures and recognition of chromosomal abnormalities with suitable software. The results of karyotype analysis here indicated that early passage iChM5Ap14 and iChM5Bp14 cells had a normal female (46, XX) karyotype without chromosomal abnormalities at a detection limit of 5 Mb (Fig. 4C). A normal karyotype was maintained in late passage iChM5Ap60 and iChM5Bp60 cells (Fig. 4C), the last passage tested. Prolonged culture of iChM5 derivatives is in contrast to the ChM5 parental cells that senesce near passage 20. Taken together, these findings show that iChM5 derivatives are karyotypically normal and self-renewing iPSCs.

### Rosette assembly in validated iChM5 derivatives

Spontaneous differentiation is expected in established PSC cultures and provides evidence for a dynamic state of pluripotency. We next asked whether loss of pluripotency gene expression could be directly associated with spontaneous rosette assembly in validated iChM5 derivatives. Immunofluorescence analysis indicated that the bulk of cells (>90%) in iChM5A and iChM5B cultures ($n \geq 3$ of each) expressed Nanog and Sox2 as well as Oct4. Dual labeling showed that nuclear localized Nanog was correlated with nuclear localized Oct4 (Fig. 5). The absence of nuclear localized Oct4 and Nanog correlated with clusters of more closely apposed cells that were reminiscent of forming neural rosettes. Immunostaining showed all of the cells tested ($n > 500$), with and without colocalized Oct4 and Nanog expression, expressed Sox2 (Fig. 5), consistent with the known maintenance of Sox2 expression during neural differentiation of PSCs. Dual labeling of Sox2 and Eg5, a well characterized kinesin that binds to cytoplasmic microtubules

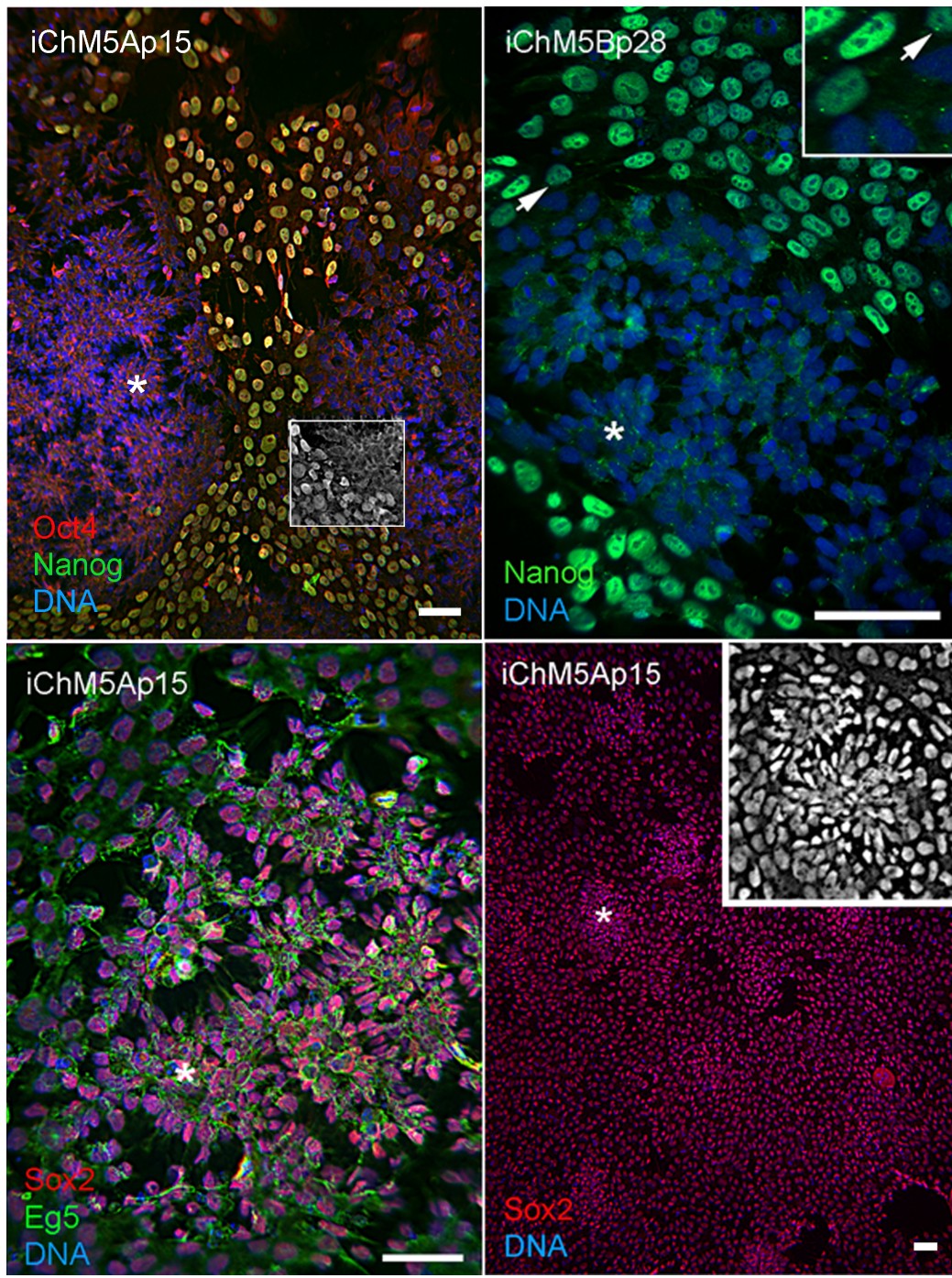

**Figure 5 Spontaneous assembly of rosettes in iChM5 derivatives.** Feeder free cultures iChM5Ap15 and iChM5Bp28 cells in chamber slides were immunostained as indicated. Left panel, emerging rosettes among self-renewing iPSCs; grayscale inset shows 1x magnification of immunostaining of Oct4 alone. Middle panel shows Nanog staining alone with inset at 2X magnification showing presumptive centrosomes (arrows). Bottom left panel shows forming rosettes immunopositive for Sox2 and Eg5. Bottom right panel shows low magnification image of immunostaining of Sox2 in this iChM5Ap15 culture, showing that virtually all cells were immunopositive. Grey scale inset shows representative forming rosette. Asterisks (*) in each panel indicates example of forming rosette. Scale bar, 50 microns.

(*Cross & McAinsh, 2014*), revealed cytoplasmic extensions that suggested changes in cell morphology during early stages of rosette assembly (Fig. 5). Screens of more than 3 fields of view in at least 3 samples of iChM5A and iChM5B and H9 cells failed to show rosette structures with nuclear co-localization of Oct4 and Nanog. These collective observations correlate coordinate loss of nuclear localized Oct4 and Nanog, but not Sox2, with the early stages of rosette assembly and validate use of rosette assembly as a sentinel for pluripotency of candidate iPSCs as well as neural differentiation potential of established iPSC lines.

During the course of these imaging experiments, we noted small equal-sized dots of Nanog immunoreactive signal at a perinuclear position in cells with and without nuclear localized Nanog (Fig. 5). The localization and regularity of these dots suggested immunoreactive signal was due to localization to centrosomes in nonmitotic cells. In an effort to test the dependence of apparent centrosome staining to Nanog expression, we transfected HEK293 cells with the 2-vector combination of episomal vectors (Table 1). Immunostaining showed nuclear localization of Nanog in a subset of cells as expected of transfected populations (Fig. 6), but immunofluorescent signal was detected at centrosomes in virtually all interphase cells, irrespective of nuclear localized Nanog expression. Although the Nanog antibody is a rabbit polyclonal antibody (Table 3) reported to be specific to phosphorlylated forms of Nanog, our results suggest recognition of cross-reactive phosphoepitopes localized at centrosomes. We have since tested for Nanog expression in parental ChM5 cells and other amniotic cell populations with similar results; amniotic cells showed centrosome localized Nanog signal, but neither nuclear localized Nanog nor Oct4 expression was detected, consistent with centrosomal staining that is unrelated to Nanog expression and pluripotency.

## Molecular analysis of iChM5 derivatives

Episomal vectors are lost when the vector encoded EBNA-1 gene is epigenetically silenced in PSCs and EBNA-1-dependent replication of episomes is blocked (*Frappier, 2012*; *Yates, Warren & Sugden, 1985*). Loss of episomes from iChM5 derivatives was evaluated with PCR, using transgene-specific primers (Table 2) to probe genomic DNA of iChM5A and iChM5B derivatives at very early (p4–6), mid (p24–25) and late (p59–60) passages and from parental ChM5p10 cells and MEFs. The EBNA-1 and OriP transgenes were detected in early, but not in later passages of candidate iChM5A and iChM5B lines (Fig. 7A), showing loss of episomal vectors during expansion of candidate populations. Detection of EBNA-1 and OriP was correlated with detection of vector transgenes in early passage iChM5p6 cultures, but not in iChM5Ap34 cultures (Fig. 7B). PCR analysis of genomic DNA and transcript analysis of 4 clonal lines derived from iChM5Ap15 indicated that episomes were lost early during culture expansion (data not shown). Taken together, these finding show recovery of vector-free iChM5 derivatives.

The results thus far showed expression of Oct4 in iChM5 derivatives, but not in the parental cells. Because demethylation of cytosines in CpG islands in the promoter of Oct4 is essential for conversion of somatic cells into self-renewing iPSCs and expression of Oct4 (*Watanabe, Yamada & Yamanaka, 2013*), methylation of CpG islands was assayed by

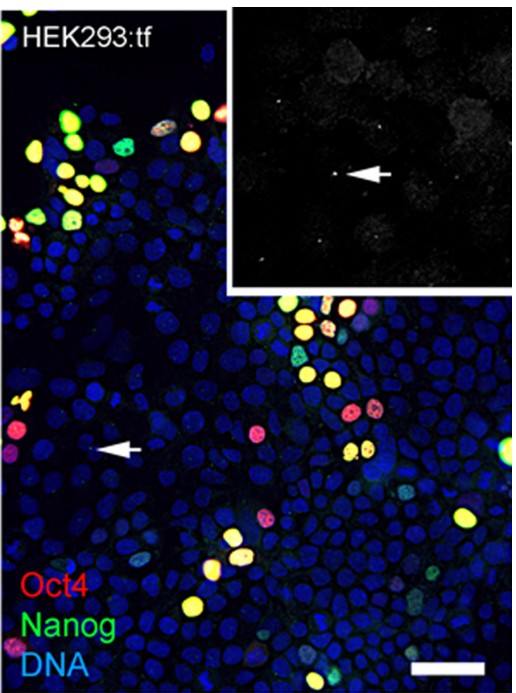

**Figure 6 Nanog signal at centrosomes in nontransfected HEK293 cells.** Image shows HEK293 cells transfected with 2-vector combination of pEP4 E02S CK2M EN2L and pEP4 E02S ET2K plasmids and stained with antibodies against Oct4 and Nanog and a fluorescent chromatin dye as indicated. Grayscale inset at 2X magnification shows Nanog signal at presumptive centrosomes (arrow) that are within focal plane of the objective. Centrosomes that are out of the focal plane are not visible here. Differential staining for Oct4 (red), Nanog (green) or Oct4 and Nanog (yellow) expression reflects the presence of Oct4 on both vectors and Nanog on one vector. Scale bar, 50 microns. This image shows that immunostaining of Nanog at centrosomes is likely independnet of Nanog expression because cells lacking Nanog localization show immunostaining of centrosomes.

bisulfite sequencing of genomic DNA from iChM5Ap18, H9p50 and parental ChM5p10 cells, using previously established primers to amplify Oct4 promoter elements (*Freberg et al., 2007*). DNA sequence analysis of cloned amplification products (Fig. 7C) showed that CpG motifs between the distal and proximal enhancers in H9p50 (9%, 4.0%) and iChM5Ap18 cells (0%, 2%), respectively, were hypomethylated relative to these motifs in parental ChM5p10 cells (43%, 31%). The segment between the proximal enhancer and the transcriptional start site showed methylation in both H9p50 and iChM5Ap18 cells (50%, 62%), respectively, that was similar to parental ChM5p10 cells (75%). These observations indicated that genetic reprogramming induced epigenetic changes in iChM5A derivatives that closely aligned with H9 hESCs. One inference of these findings is that epigenetic silencing underlies the lack of Oct4 expression in parental ChM5 cells and that immunodetection of Oct4 and other pluripotency genes in iChM5A and iChM5B lines reflects epigenetic modifications of chromatin that allow transcription.

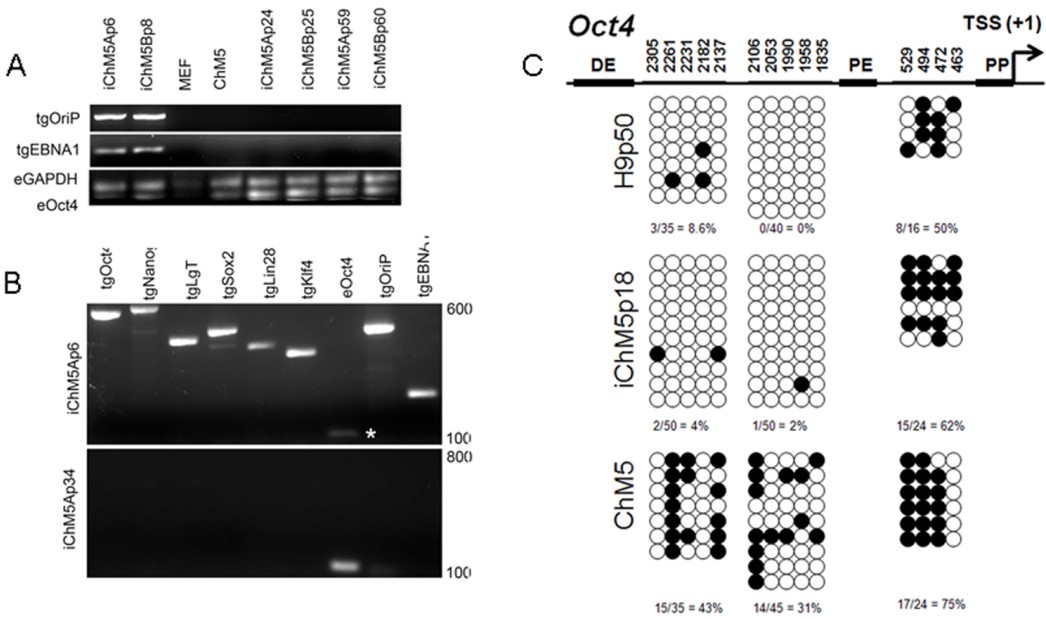

**Figure 7 Molecular analysis of iChM5 and iChM5B lines.** (A) Amplification of vector sequences. Genomic DNA was probed for the vector transgenes (tg) OriP and EBNA-1 and for endogenous (e) GAPDH and Oct4 genes as controls. Note that the eOct4 band is near the gel edge. (B) Amplification of transgenes. Genomic DNA from iChM5Ap34 and iChM5Ap6 cells was probed for the vector transgenes Oct4, Nanog, SV40 T-antigen, Sox2, Lin28, Klf4 and the endogenous copy of Oct4. The upper and lower range of ladder markers are indicated. Data in (A) and (B) show that vectors are lost from iChM5 derivatives during population expansion. (C). Bisulfite sequence analysis of methylation of Oct4 promoters elements. Diagram shows the Oct4 promoter containing a distal enhancer (DE), proximal enhancer (PE), proximal promoter element (PP) and transcription start site (TSS + 1). Open and closed circles represent unmethylated and methylated cytosines, respectively, at the positions indicated as inferred from DNA sequence analysis of cloned fragments. Each row of circles represents the cytosines in CpG motifs from a single cloned fragment. The percentage of methylated cytosines in each clone set is indicated. These results show that methylation of the parental cells was modied during reprogramming and became similar to that of control H9 hESCs.

## Transcript profiles

Transcription of Oct4 and other genes in the pluripotency network was tested directly by syber green-based quantitative amplification of cDNA (Fig. 8A). Transcripts of Oct4, Sox2, Nanog and Lin28 were not detected above internal controls in cDNA from parental ChM5 cells, but were detected in iChM5A and iChM5B cells and in H9 hESCs. Transcript levels of cMyc were above internal controls in parental ChM5 cells and were similar to levels in iChM5 derivatives and control hESCs although trending lower. Variation in transcript levels was expected given the potential for differentiation within populations, but Sox2 levels were unexpectedly low. The activity of our Sox2 primers were tested by transcript analysis of cDNA generated from a commercial source of immortalized NSPs derived from human fetal ventral midbrain. The results showed down regulation Oct4 and Nanog, but up regulation of Sox2. Transcript levels in iChM5-derived NSPs were similar, but up regulation of Sox2 was less dramatic (data not shown). These findings indicated that the low Sox2 levels in iChM5 candidates and H9 hESCs did not reflect the Sox2 primers, but

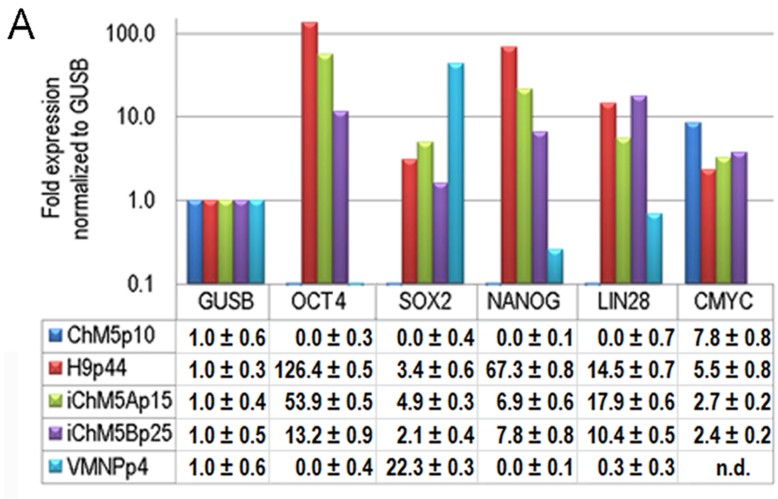

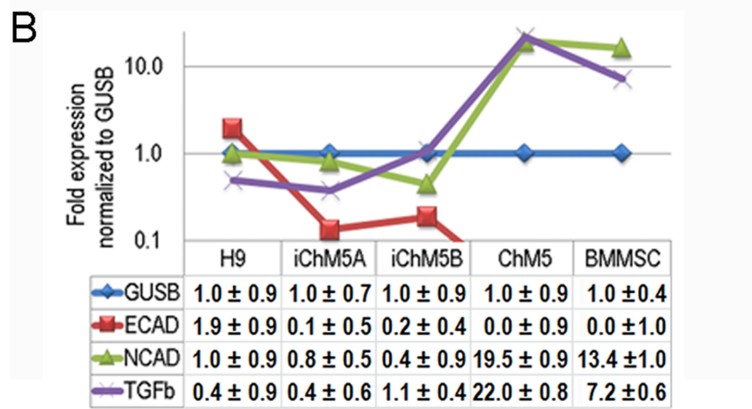

**Figure 8  Transcript profiles.** (A) Transcript profiles of pluripotency factors. ddCt values for ChM5p10, H9p44, iChM5Ap18, iChM5Bp20 cells and immortalized human ventral midbrain neural progenitors (hVMNSPs) were normalized to levels of b-glucuronidase (GUSB). cMyc levels in single experiment indicated with asterisks (*) or not determined (n.d.). (B) Transcript profiles of EMT-associated genes. ddCt values for H9p44, iChM5Ap15, iChM5Bp37, ChM5Ap10, BMMSCp5 were probed for GUSB, E-Cadherin (ECAD), N-Cadherin (NCAD) and TGF$\beta$ with TaqMan gene expression assays and presented as fold expression as normalized to GUSB.

the relative levels of Sox2 transcripts in these PSC cultures. Taken together, these results confirm transcriptional activation of the pluripotency network in iChM5 derivatives.

Somatic cell identity is lost or down regulated during genetic reprogramming. Although the somatic source(s) of the parental ChM5 mixed cell pool is unknown and cannot be tested directly, amniotic stromal and epithelial cells alike show stromal cell traits (*Wilson et al., 2012*). Stromal cell traits of amniotic epithelial cells can reflect epithelial-mesenchymal transition (EMT) in which epithelial cells acquire stromal cell traits by down regulation of E-Cadherin and up regulation of N-Cadherin (*Nieto, 2011*). TaqMan assays were used to probe transcript levels of these cadherins and the EMT inducer TGF$\beta$ in cDNA from parental ChM5 cells, iChM5 derivatives, H9 hESCs and bone marrow derived stromal cells (BMMSCs) as a stromal cell control (Fig. 8B). Transcript analysis showed 10 fold lower

levels of N-Cadherin and TGF$\beta$ in iChM5 derivatives in comparison to parental ChM5 cells and BMMSCs. E-Cadherin was undetected in parental ChM5 cells and BMMSCs, but a low level was detected in iChM5 derivatives although lower than levels in H9 hESCs. These findings together show loss of stromal cell characteristics by genetic reprogramming of parental ChM5 cells.

## DISCUSSION

The promise of genetic reprogramming has prompted initiatives to develop banks of induced pluripotent stem cells (iPSCs) from diverse sources, in part because immuno-logically compatible iPSCs from allogenic sources is the more likely path for clinical applications (*Turner et al., 2013*). Participation of diverse research groups in development of iPSC lines and technologies will benefit from methods that differentiate between pluripotent developmental potential and partially reprogrammed candidates or simply cells expressing pluripotency genes without pluripotent developmental potential. Here, we targeted human amniotic cell populations that we generated in a previous work (*Wilson et al., 2012*) and report a novel use of neural rosettes as a sentinel for induced pluripotency in candidate iPSC lines and in validated PSC lines.

### Self-assembly of neural rosettes as a sentinel for induced pluripotency

Neural rosettes represent a 3-deminisional primitive tissue that approximates the primordial neural tube *in vivo* (*Elkabetz & Studer, 2008*; *Wilson & Stice, 2006*). Spontaneous self-assembly of neural rosettes is unique to PSCs and rosette structures in teratomas are commonly cited as evidence of neural differentiation potential of PSCs. Derivation of neural rosettes has been used primarily to generate cultures of NSPs from PSCs (*Ebert et al., 2013*; *Shin et al., 2006*; *Yan et al., 2013*) or to study signaling pathways in specification of neural subtypes (*Chambers et al., 2009*), but use of rosette assembly has not been reported in the literature as a means to screen and select candidates for expansion and validation. Progression through a rosette stage is not essential for directed transdifferentiation of somatic cells into neural derivatives (*Ladewig, Koch & Brustle, 2013*), but self-assembly of neural rosettes is arguably an essential capacity of PSCs and provides a measure of confidence in candidate selection.

Rosette assembly has practical value in candidate selection for several reasons. First, rosette assembly can occur by spontaneous differentiation of candidates without application of neural induction protocols. Second, the 3-dimensional structure and organization of rosettes can be readily identified in living cultures by phase imaging and distinguished of from aging MEFs, parental cells and amorphous cell aggregates. Third, spontaneous differentiation of rosettes generates a diverse array of derivative cell types that can be validated by immunostaining of nuclear localized transcription factors (*Elkabetz & Studer, 2008*; *Wilson & Stice, 2006*) and use of dual labeling of different transcription factors to enhance the rigor of the assay. This is a key advantage because nuclear localized transcription factors are superior indicators of neural identity in comparison to more widely used cytoplasmic markers such as nestin and $\beta$III-tubulin that in our hands are sensitive to

technical artifacts in fixation and immunostaining. Finally, functional tests are less likely to give false positives in comparison to marker expression alone. Expression of pluripotency markers does not guarantee pluripotency; established hESC lines harboring chromosomal abnormalities can express pluripotency factors, but fail to differentiate (*Wilson et al., 2007*) and integrated transgenes may not be fully silenced (*Malik & Rao, 2013*; *Mostoslavsky, 2012*; *Rao & Malik, 2012*) and mistaken for expression of endogenous genes.

## Activation and inactivation of the pluripotency network in iChM5A and iChM5B lines

The value of neural rosettes in candidate selection was substantiated by subsequent validation of pluripotency of iChM5 derivatives, including evidence for epigenetic modification of chromatin structure (Fig. 7C) as expected of activation of the endogenous pluripotency network of genes (Fig. 8A) and down regulation of stromal cell characteristics of parental ChM5 cells (Fig. 8B). Pluripotency is a dynamic state that is difficult to convey in static images, but evidence is critical to discerning differences between expression of pluripotency genes and pluripotent differentiation potential. Here, a dynamic state of pluripotency was evident in spontaneous assembly of neural rosettes in cultures of validated self-renewing iChM5 derivatives; loss of nuclear localized Oct4 and Nanog correlated with changes in cell morphology in forming neural rosettes (Fig. 5). This immunofluorescence assay is valuable because it is simple, highly reproducible ($n \geq 6$) and can provide critical internal controls in the same culture and within the same field of view. Immunostaining in this case is superior to flow cytometry that cannot discriminate between nuclear and cytoplasmic localization of transcription factors or easily correlate gene expression and changes in cell morphology in differentiating cells.

Teratoma formation is the accepted standard for pluripotent developmental potential and an assay for the safety of iPSC derivatives in clinical applications (*Muller et al., 2010*). iChM5A and iChM5B derivatives generated teratomas, under the same conditions and within the same timeframe as control H9 hESCs (Fig. 4B). We used VPA during reprogramming of ChM5 cells; VPA is a small molecule inhibitor of histone deacetyltransferases (HDACs) that is widely used in combination with reprogramming factors in the form of transgenes, mRNA or proteins to promote reprogramming (*Huangfu et al., 2008*). Subsets of amniotic cells that were selected for expression of the cKit cell surface receptor, cultured in conditions for hESCs and transiently exposed to VPA showed characteristics of pluripotency, including tumor formation *in vivo* (*Moschidou et al., 2012*). Here we ascribe induced pluripotency of iChM5 derivatives to genetic reprogramming rather than chemical induction by VPA because newly isolated candidates contained episomal vector sequences (Figs. 7A and 7B) and because VPA produces global effects on transcription levels that are not known to be heritable.

The value of teratomas as assays for pluripotency is under discussion (*Buta et al., 2013*), in part because evaluation of teratoma composition has a subjective component and standards for assigning tissue derivatives could vary among research groups. We favor use of reliable organoid assays in vitro, such as neural rosettes, to characterize differentiation because such assays and their interpretation are more transparent to researchers and

because of the availability of rigorous internal controls. That said, reliable methods to generate organoids from nonneural lineages are only recently beginning to appear in the literature and are not yet broadly used as criteria for differentiation potential.

## Differential capacity of iChM5 derivatives for rosette assembly

We show that rosette assembly distinguished iChM5 candidates from iChM5RCB1 candidates. Neural rosettes formed in backup cultures of iChM5A and iChM5B candidates that were comparable to rosettes in H9 controls (Fig. 3, Figs. S1 and S2). Rosettes were not detected in backup cultures of iChMRC.B1-derived candidates although these candidates were generated by transfection with the same 3 vector combination that produced iChM5B candidates. Similar results were obtained by transfection of the ChM1 population (data not shown) that is highly enriched for epithelial cells (*Wilson et al., 2012*). The simplest interpretation of these findings is that amniotic stromal cells are easier to reprogram with episomal vectors than epithelial cells and that differences in reprogramming efficiency is reflected in the differential capacity of the candidates to assemble neural rosettes.

The underlying cause of differential reprogramming here is uncertain. We show (Fig. 1) that the episomal vectors are unstable in HEK293 cells and we infer that maintenance of all 3 vectors in any one cell is a very low probability event. We are unaware of data reporting selective retention of different episomal vectors, but expression of cMyc on the smallest episomal plasmid in the 3-vector combination (Table 1) could confer some selective advantage in maintenance over the other 2 plasmids. If so, the reduction of Oct4 copy number would be detrimental to reprogramming as others show that one copy of an episomal vector encoding Oct4 was insufficient to reprogram fetal fibroblasts (*Yu et al., 2009*). Vector systems and reprogramming protocols have improved since we initiated this work and further work could show whether the differences in reprogramming reflect reprogramming methods or differences between epithelial and stromal cell types in amniotic fluid or from other sources.

Our findings beg the question of whether rosette assembly is a universally valid sentinel of pluripotency. Universality cannot be tested, but several observations are consistent with this expectation. First, rosette assembly reflects differentiation of PSCs to neuroepithelial cells and assembly of cell:cell adhesions and junctions that closely align with assembly of neuroepithelia of the primordial neural tube (*Elkabetz & Studer, 2008*; *Wilson & Stice, 2006*). Second, given that differentiation of neuroectoderm is the first germ layer lineage generated during gastrulation of embryos and neural differentiation is an essential benchmark of pluripotency, it is reasonable to expect spontaneous assembly of neuroepithelia in the form of neural rosettes in PSCs. Importantly, assembly of rosettes does not necessarily translate into recovery of PSCs since pluripotency can be lost for a variety of reasons, including differentiation of PSCs in the absence of PSC self-renewal. Third, we show here that rosettes form in synthetic media under feeder free conditions (Fig. 5), indicating that rosette assembly is not media-dependent. Several explanations can be offered for the lack of rosette assembly in candidate lines and other putative PSC lines. First, the candidates, like iChMRC.B1 candidates, are not pluripotent. Second, the

candidates/PSCs may harbor chromosomal abnormalities (*Wilson et al., 2007*), blocking differentiation in spite of Oct4 expression. Third, they form, but they are overlooked and/or not recognized as rosettes. Finally, rosette assembly in PSCs derived from rodent and other species has not been emphasized the literature. It is unclear if this is an oversight, culture conditions using widespread use of passaging with trpsin or for other reasons. Ultimately, the choice to use rosettes as a sentinel is left with the research group. We considered rosette assembly as an essential benchmark of pluripotency that increases rigor in derivations of iPSC lines and expand resources for further research.

## Conclusions and repository access

We show recovery of vector-free fully validated iPSCs by genetic reprogramming of cells derived from amniotic fluid with episomal vectors. Spontaneous assembly of neural rosettes provided a sentinel for candidate selection in advance of validation. Coordinated loss of nuclear localized Oct4 and Nanog in emerging neural rosettes in cultures of self-renewing iPSCs provides a simple and reliable assay for a dynamic state of pluripotency to differentiate pluripotent developmental potential of PSCs from expression of pluripotency genes in somatic cells. Rosette assembly and differentiation is not new to stem cell research, but could maximize resource allocation in derivation and use of PSCs and improve the quality and quantity of iPSCs from diverse sources for clinical applications.

The lines generated in this work are available as PGW1i:ChM5A and PGW2i:ChM5B on request from the Rutgers University Cell and DNA Repository, 145 Bevier Road Piscataway NJ 08854-8009.

## ACKNOWLEDGEMENTS

We are grateful to Samantha Jeschonek who generated Fig. S3 while a Summer Scholar at WFIRM. We are grateful to Cynthia Zimmerman, Laddie Crisp, Cathy Mathies, and Raymond Johnson for technical support and insight. We appreciate the thoughtful comments and discussion with members of WFIRM.

### Funding

The authors received funding support from the Christopher L. Mosley Foundation (PGW), Telemedicine & Advanced Technology Research Center (W81XWH0710718) and the State of North Carolina (G20431003411MED). The funders had no role in study design, data collection and analysis, decision to publish, or preparation of the manuscript.

### Grant Disclosures

The following grant information was disclosed by the authors:
Christopher L. Mosley Foundation (PGW).
Telemedicine & Advanced Technology Research Center: W81XWH0710718.
State of North Carolina: G20431003411MED.

## Competing Interests

The authors declare there are no competing interests.

## Author Contributions

- Patricia G. Wilson conceived and designed the experiments, performed the experiments, analyzed the data, contributed reagents/materials/analysis tools, wrote the paper, prepared figures and/or tables, reviewed drafts of the paper.
- Tiffany Payne performed the experiments, analyzed the data.

## Human Ethics

The following information was supplied relating to ethical approvals (i.e., approving body and any reference numbers):

Wake Forest Health Sciences Institutional Review Board: IRB#00007486.

## DNA Deposition

The following information was supplied regarding the deposition of DNA sequences:

Figshare http://dx.doi.org/10.6084/m9.figshare.1153969.

## Supplemental Information

Supplemental information for this article can be found online at http://dx.doi.org/10.7717/peerj.668#supplemental-information.

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
