# Peer review of "Genetic reprogramming of human amniotic cells with episomal vectors: neural rosettes as sentinels in candidate selection for validation assays"

_PeerJ, doi:10.7717/peerj.668_

## Round 0.1 · original submission · Major Revisions

· Academic Editor

Major Revisions

Thank you for submitting your manuscript to PeerJ. After careful consideration, we feel that it has merit, but it isn't suitable for publication as it currently stands. Therefore, my decision is "Major Revision". Please address the reviewers' comments.

Reviewer 1 ·

Basic reporting

This article is well designed with experimental work. I recommend to accept this paper with minor revisions.

Experimental design

well designed with good reporting.

1. In figure 4 D (transcript analysis) number of experiments performed (N?) is missing and error bars?

2. Need to be further data and better explanation of the passage number and nomenclature.

3.As authors claim that there cells are induced pluripotent stem cells (iPSCs), do they have observed differentiation of these cells in other cells types than the "Neural cells"

4. Why the optimal colonies were pooled?

5. Why Rosettes did not appear in backup cultures of clone "iChMRCB1" and where as they appeared in "back up cultures of iChM5A and iChM5B" ? any explanation

6. Better explanation of Figure 3 "Karyotype analysis"

Validity of the findings

acceptable

Reviewer 2 ·

Basic reporting

In this MS, the authors has used “rosettes as sentinels in candidate selection for validation assays” and performed a series of microscopy as well as gene expression studies. The outcome is a generation of two cell lines, namely iCh5A and iChM5B which they made available for others as well (on request). The work is a straight. However, there are few issues that need to be addressed before it can be considered for acceptance.
1. Almost the entire MS lacks the use of negative controls, especially in case of antibody stainings. It is important to demonstrate the specificity of these antibodies also. I could not find the antibody details such as the of the antibody sources, dilutions etc. These information must be added. How can one be sure that these staining’s are specific?
2. It is very difficult to comment on the level of expression based on the antibody staining.
3. Authors use transfection and efficiency is mentioned as low. Under these condition, there will be a lot of non-transformed cells. Authors must clarify how they select the transfected one.

Experimental design

Need to include positive and negative controls where applicable. Specificity of the antibodies need to be included.

Validity of the findings

No comments

Comments for the author

From the MS, it is not very clear how this cell lines and the findings will help further research. Authors must add these in the discussion section. Also whether the reproducibility of such key findings hold true in all conditions and cases, need to be discussed. May be formation of rosettes as sentinels in candidate selection for validation assays is not universal in nature. This needs to be discussed.

Are these cell lines are from Male or from female? This is relevant as karyotyping was done. It will be better if the authors mark the individual chromosomes also.

---

## Round 0.2 · accepted · Accept

· Academic Editor

Accept

Given the positive reviews and the reasonable revisions to address each query, I do not see any reason to delay the paper by sending it back out for additional review. I am happy to accept your manuscript; congratulations.